# Electrochemical Corrosion of SAC Alloys: A Review

**Ali Gharaibeh [1], Ilona Felhősi [2], Zsófia Keresztes [2], Gábor Harsányi [1] , Balázs Illés [1] and Bálint Medgyes [1],***

[1] Department of Electronics Technology, Budapest University of Technology and Economics, 1111 Budapest, Hungary; aligharaibeh@ett.bme.hu (A.G.); harsanyi@ett.bme.hu (G.H.); billes@ett.bme.hu (B.I.)

[2] Institute of Materials and Environmental Chemistry, Research Centre for Natural Sciences, 1117 Budapest, Hungary; felhosi.ilona@ttk.hu (I.F.); keresztes.zsofia@ttk.hu (Z.K.)

* Correspondence: medgyes@ett.bme.hu; Tel.: +36-1-463-2105

**Abstract:** Tin–silver–copper (SAC) solder alloys are the most promising candidates to replace Sn–Pb solder alloys. However, their application is still facing several challenges; one example is the electrochemical corrosion behaviour, which imposes a risk to electronics reliability. Numerous investigations have been carried out to evaluate the corrosion performance of SAC lead-free alloys, regarding the effect of the corrosive environment, the different manufacturing technologies, the effect of fluxes, the metallic contents within the SAC alloys themselves, and the different alloying elements. In these studies, widely used electrochemical techniques are applied as accelerated corrosion tests, such as linear sweep voltammetry and electrochemical impedance spectroscopy. However, there is lack of studies that try to summarise the various corrosion results in terms of lead-free solder alloys including low-Ag and composite solders. This study aims to review these studies by showing the most important highlights regarding the corrosion processes and the possible future developments.

**Keywords:** electrochemical corrosion; SAC alloys; linear sweep voltammetry; electrochemical impedance spectroscopy

## 1. Introduction

For decades, lead-bearing solders (Sn-Pb) have been widely used in electronics manufacturing due to their solderability behaviours such as good mechanical strength, excellent surface wetting behaviour, low melting point, good corrosion resistance, and low cost [1–3]. However, lead as a heavy metal was banned for soldering [4]. Therefore, different lead-free solder alloys have been developed and investigated [5,6]. Among the lead-free solder alloy types, Sn-Ag-Cu (SAC) alloys were the most promising candidates to replace Sn-Pb solder alloys because of their beneficial characteristics such as relatively low cost, low melting temperature, and advantageous thermal and mechanical properties [7–10] compared to other lead-free ones. However, for reliable operation, the SAC alloys have to meet also many challenges [11]. One of them is the electrochemical corrosion (ECC), which can impose very high risk on the long-term application of the SAC lead-free solder joints.

The ECC for the metallic materials referred to the dissolution of metal and the degradation of metal properties as a function of time [12]; in the presence of electrolyte, the ECC requires two or more reactions on a metal surface. One of the fundamental reactions must involve metal atoms losing electrons (e.g., oxidation), and the metal in this case is called the anode, and another reaction must involve metal ions or atoms gaining electrons (e.g., reduction) and the metal, in this case, is called the cathode. ECC occurs when there is a flow of electric current in the electrolyte from the anode to the cathode.

There are several types of ECC such as uniform corrosion [13], galvanic corrosion [14], pitting corrosion [15], and crevice corrosion [16]. Several studies were conducted to highlight the harmful effect of this phenomena on the different practical applications such as High-Voltage Direct Current (HVDC) transmission lines [17], electric energy meters [18], and dentistry field [19,20].

SAC alloys, as replacement of the Sn-Pb solders used in electronics, have been proposed in three different countries with different compositions (typically): Europe (Sn-3.8Ag-0.7Cu), USA (Sn-3.9Ag-0.6Cu), and Japan (Sn-3.0Ag-0.5Cu) [21]. Although SAC alloys already showed some results related to solderability enhancement, further improvement regarding the microstructure and interface were done by the addition of other alloying elements such as cobalt [22], bismuth [23], cerium and lanthanum [24], titanium [25], zinc [26], indium [27], and manganese [28]. In addition to adding the alloying elements, alloying with nanoparticles was found to enhance SAC alloys' microstructure, shear strength, and hardness such as $Al_2O_3$ nanoparticles [29], Mo nanoparticles [30], $SrTiO_3$ nanoparticles [31], $TiO_2$ nanoparticles [32], and $ZrO_2$ nanoparticles [33]. The main challenge for the alloying elements and nanoparticles to SAC alloys is to determine if they enhance SAC alloys' properties and to determine the weight percentage that makes the optimum improvement among the suggested additions.

A new tendency in the development of the SAC solder alloys is to decrease the silver content. The so-called "Lead-free Micro-alloyed low Ag Solder Alloys" are suggested as a development to the traditionally SAC alloys [34,35], where the Ag content in these alloys is much smaller (under 1 wt %), and the alloys have to be candidate with micro-alloying (in range of 0.05 to 0.15%) elements such as nickel, bismuth, cobalt, antimony, etc. Improvement could be achieved by reducing the silver content in the alloy. On the one hand, it will reduce the possibility of the formation of $Ag_3Sn$ intermetallic compound (IMC) particles in the solder joints, which degrade the mechanical parameters. On the other hand, the price of micro-alloyed SAC is 5–10% cheaper than the traditional SAC alloy.

Since the conversion from Sn-Pb to lead-free solders, the usage of no-clean flux became dominant in the soldering, and due to the relative higher melting point of SAC alloys compared to Sn-Pb alloys [36], their implementation brought a more activated no-clean flux to remove the existed metal oxides from the soldered parts and improve the solderability. The improper use of no-clean flux resulted in more flux residues, which may cause defects under the action of humidity and resulted in failure in the electronics. Therefore, the soldering process should be followed by a suitable cleaning process.

Reviews on the corrosion characterisation of Sn-Zn alloys [37] and the corrosion performance of lead-free solder alloys are already published [38–40]. However, these reviews do not cover corrosion information related to micro-alloyed low Ag solder alloys and the variation of the elemental composition within SAC solder alloys. Our work reviews and evaluate the results about the corrosion behaviour of SAC alloys related to different compositions and alloying elements, different kinds of corrosive environments, and applied test methods.

## 2. Techniques to Study Electrochemical Corrosion (ECC)

### 2.1. Linear Sweep Voltammetry (LSV)

LSV is one of the most widely used methods for the corrosion resistance evaluation [41–46]; it is suitable to determine the corrosion rate and to study the kinetics and mechanism of corrosion processes. The sample is usually scanned in a large potential range with a sufficiently slow fixed scan rate to get a steady-state polarisation curve [41]. In each region of the polarization curves, we obtain information about the electrochemical processes taking place. In the cathodic region of the polarisation curves, usually, the reduction of dissolved oxygen takes place according to the following equation [47–49]:

$$O_2 + 2H_2O + 4e^- = 4OH^-,$$

which is the most common rate-determining cathodic reaction when the corrosion of metals occurs in near-neutral solutions containing dissolved oxygen. Near the corrosion potential ($E_{corr}$), the corrosion

current density ($i_{corr}$) can be determined most often with the Tafel-extrapolation method [50–52]. In the anodic part of polarisation curves, several regions can be distinguished: metal dissolution, passive film formation, and a transpassive dissolution range with possible anodic reactions. Usually, the surface of solder joints is covered with a thin oxide layer, which protects the underlying metal from corrosion.

The typical polarisation behaviour of SAC solder alloy in NaCl solution is visible in Figure 1, where four different potential ranges in the anodic part can be distinguished, illustrating the most important corrosion characteristics. On scanning from point B in the anodic direction, there is an increase in the current density due to active dissolution of the metal species in the solder alloy; usually, the metals with a lower standard electrode potential have the priority in the dissolution reaction, the active dissolution continues until the concentration of dissolved metal reaches a maximum current density at point C, which is referred to as the critical current density ($i_{crit}$), and the potential is referred to as the passivation potential ($E_p$). On further scanning from point C, the formation of a passive film occurs as the solid oxide precipitates on the electrode surface, which is reflected in a decrease in the current density up to point D. Here, the current density is found to be independent of the potential; the current density at point D is referred to as the passivation current density ($i_p$), and the potential range where the current density remains constant is referred to as the passivation range. Beyond point E, the alloy runs into a transpassive region (EF) where the current density increases, indicating the breakdown of the passive oxide layer due to the presence of $Cl^-$ on the surface. The previous explanation is consistent with the findings in [53,54]. However, all of the typical regions do not need to occur in the corrosion studies, since not all SAC alloys show the passivation behaviour [49,52].

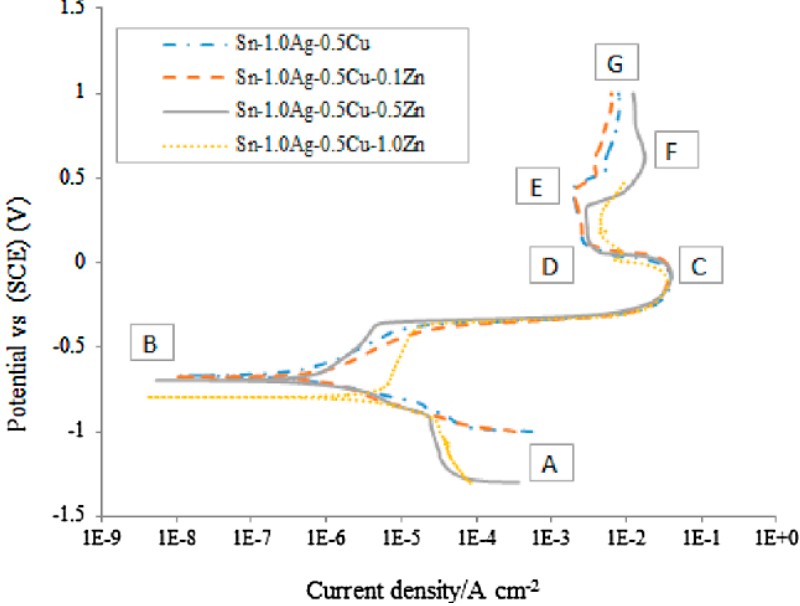

**Figure 1.** Potentiodynamic polarisation curve of SAC105-XZn (*X* = 0, 0.1, 0.5, 1 wt %) solder alloys in 3.5 wt % NaCl solution [55].

The higher $E_{corr}$ indicates the higher required driving force to initiate the corrosion, but it is not recommended to evaluate the overall corrosion resistance based only on this value, since it is affected by many factors. Instead, $i_{corr}$ is generally used to decide on the corrosion resistance [48,56–58]. Another criterion is presented in [43,53,54], in which a better corrosion resistance is associated with the larger passivation range and with the lower passivation current density ($i_p$). Wahida et al. [59] reported three quantities that are used to evaluate the corrosion: $\Delta E_1 = E_{break} - E_{corr}$, which expresses pitting susceptibility, $\Delta E_2 = E_{break} - E_p$ as re-passivation ability, and $\Delta E_3 = E_p - E_{corr}$ as the probability of localised corrosion. However, LSV is not suitable for evaluating long-term corrosion performance, since during polarisation, the electrochemical processes take place with an enhanced rate.

### 2.2. Electrochemical Impedance Spectroscopy (EIS)

EIS is most often measured at 0 V bias (at open circuit potential) in order to determine the corrosion rate from the polarisation resistance. The impedance measurement at the open circuit potential can also be used to monitor corrosion in time, as the amplitude used is small enough not to significantly affect corrosion. On the other hand, EIS measurements at different bias potentials are also of interest when the kinetics of electrochemical processes taking place at a given potential is the subject of interest. According to our knowledge, these kinds of studies have not been deeply addressed in the literature of SAC alloys.

EIS measurements are performed by applying alternating current (AC) with a small amplitude (usually 10–15 mV) and a wide frequency range to an electrochemical cell and by measuring the response current [56,57,59–63]. The impedance is expressed by $Z(\omega) = Z_0 (\cos(\varphi) + j\sin(\varphi))$. The real part is usually plotted on the *X*-axis, and the imaginary part is plotted on the *Y*-axis to produce a Nyquist plot, which is generally characterised by a semicircle or capacitive loop from high to low frequency. The diameter of the semicircle indicates the magnitude of polarisation resistance ($R_p$) [52,56,63]. Figure 2 illustrates the Nyquist plot concept by showing some impedance spectra measured on SAC alloys.

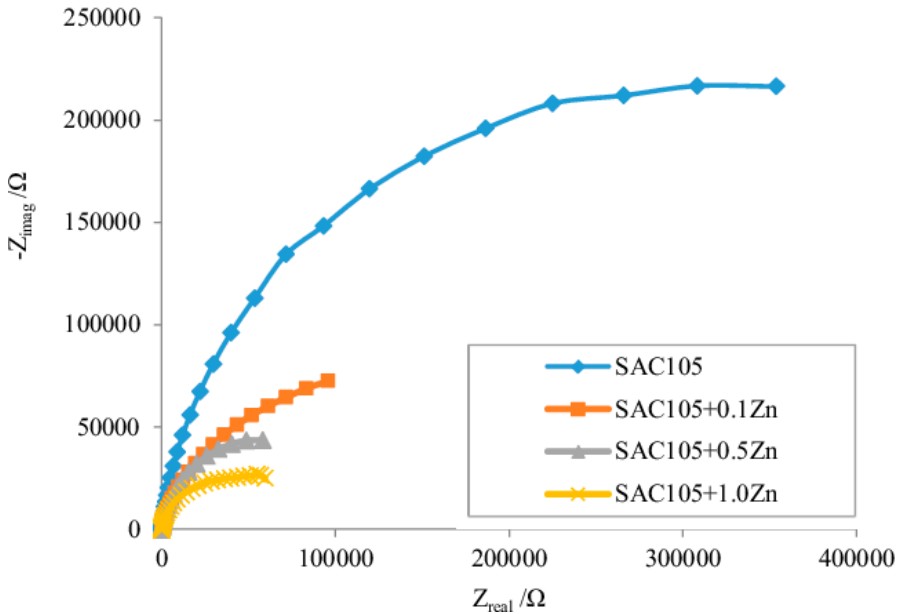

**Figure 2.** Nyquist plot for SAC105-*X*Zn (*X* = 0, 0.1, 0.5, 1 wt %) solder alloys in 3.5 wt % NaCl solution [55].

Another representation of the EIS measurement is the magnitude and phase angle on the Bode plot, as shown in Figure 3. Both values are used to evaluate the corrosion resistance, the higher magnitude of impedance (|Z|) (Figure 3a) and higher maximum phase angle (Figure 3b) indicates better corrosion resistance. For example, from Figures 2 and 3, it can be seen that the increase with the addition of Zn reduces the corrosion resistance of SAC105 alloy in 3.5 wt % NaCl solution.

EIS showed proficiency in the evaluation of pitting corrosion under atmospheric environments; it was found that in case of wet–dry cyclic conditions in an NaCl environment [64], the preferable occurrence of pit generation and growth came only when there was an existence of a very thin film of water on the specimen surface, while the preferable occurrence of re-passivation was when the surface dried out during subsequent immersion, following the complete drying of the surface.

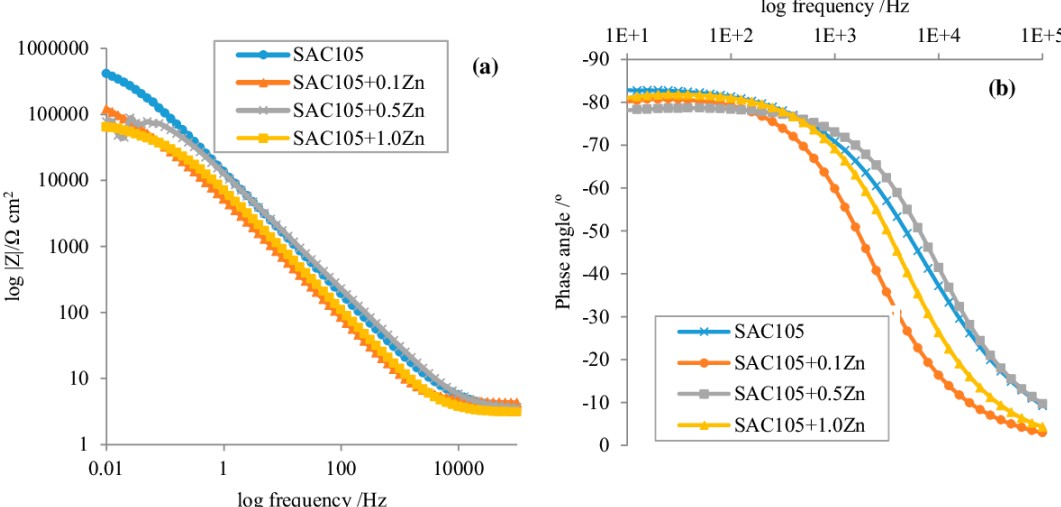

**Figure 3.** Bode plots: (**a**) Magnitude Bode plot; (**b**) Phase angle Bode plot for SAC105-XZn (X = 0, 0.1, 0.5, 1 wt %) solder alloys in 3.5 wt % NaCl solution [55].

Due to the complex nature of corrosion of SAC alloys, there are several equivalent circuits in the literature used to fit EIS data [43,60,65,66]; the most often used equivalent circuits are presented in Figure 4. Figure 4a no significant amount of oxide layer is present on the surface and/or the oxide layer does not affect the impedance response. Figure 4b the surface is covered with a dense passive oxide layer. Figure 4c the surface is covered with porous corrosion products. Figure 4d the surface is covered with a porous oxide layer, and the charge transfer of metal dissolution is diffusion controlled.

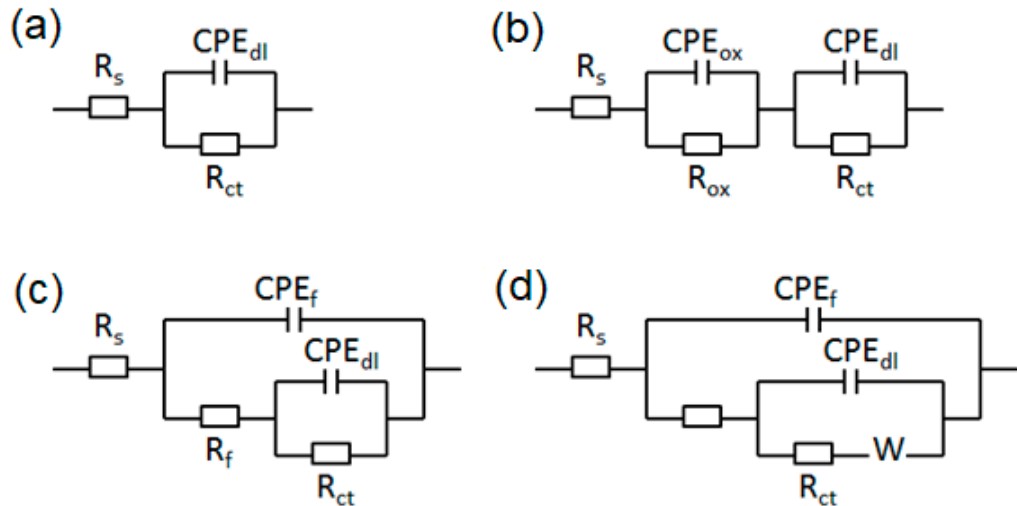

**Figure 4.** Different equivalent circuits most often used to fit EIS data of SAC alloy corrosion: (**a**) oxide layer not influence the impedance response; (**b**) surface is covered with a barrier oxide layer; (**c**) surface is covered with a porous oxide; (**d**) the surface is covered with a porous oxide and the charge transfer of metal dissolution is diffusion controlled. $R_s$: solution resistance, $CPE_{dl}$: Constant Phase Element (*CPE*) of double layer, $R_{ct}$: charge transfer resistance of metal dissolution, $CPE_{ox}$: CPE of passive oxide layer, $R_{ox}$: resistance of passive oxide layer, $CPE_f$: CPE of porous surface film, $R_f$: resistance of electrolyte in the pores of surface film, *W*: Wartburg element of diffusion.

The corrosion rate is determined from the reciprocal of the overall corrosion resistance ($R_t = R_f + R_{ct}$) [67,68]. From the microelectronics reliability point of view, the corrosion rate should be as low as possible; the acceptable corrosion rate of less than 0.1 mm/year was suggested in 0.1 M NaCl aqueous solution [69] to ensure the service life of components and devices.

LSV and EIS are the most widely used techniques for evaluating the corrosion behaviour of SAC alloys; other techniques such as Electrochemical Noise (EN) are also applied rarely [70], where pitting corrosion on the electrode surface is reflected by a fluctuation of random high vibration of the amplitude of current noise and potential noise.

## 3. Main Corrosion Mechanism of SAC Alloys in Different Environments

The corrosion resistance is a very important property in electronics reliability. Table 1 shows the electrode potentials of the typical elements in solder alloys.

**Table 1.** Reactions and standard electrode potentials of the elements in tin–silver–copper (SAC) solders [71].

| Element | Half-Reaction | Standard Electrode Potential (V) |
|---------|---------------|----------------------------------|
| Tin | $Sn^{2+} + 2e^- = Sn$ | −0.1375 |
| Silver | $Ag^+ + e^- = Ag$ | +0.7996 |
| Copper | $Cu^+ + e^- = Cu$ | +0.5210 |
| - | $Cu^{2+} + 2e^- = Cu$ | +0.3419 |
| Lead | $Pb^{2+} + 2e^- = Pb$ | −0.1262 |

In Table 1, the close electrode potentials for Sn and Pb indicate the good galvanic corrosion resistance for Sn-Pb alloys. However, in the case of SAC alloys, the much higher value of the electrode potential of Ag may result in the dissolution of Sn into the corrosive solution. The presence of intermetallic compounds (IMCs) such as $Ag_3Sn$ and $Cu_6Sn_5$ also plays an important role in the corrosion process. Since the $Ag_3Sn$ has a similar electrode potential as Ag, it is expected for the large $Ag_3Sn$ IMC particles to act as an artificial cathode that promotes the dissolution of Sn from the matrix [62,72–74]. However, it has been found also that the existence of fine and homogenously distributed IMCs represent protection against the corrosion [41,53].

To enhance the corrosion resistance of SAC solder alloys, their microstructure should be refined. Many investigations have already been done to achieve this goal. Most of the investigations focus on the NaCl electrolyte environment, since it is the most common contamination with many possible sources, such as the dust in the air, salt spray from the seasides, human sweat and fingerprints, and from flux residues during manufacturing processes [75].

### 3.1. Corrosion Studies of Pure SAC Alloys

The corrosion of pure SAC alloys could be investigated with the variation of Ag and Cu content; in aerated 3.5 wt % NaCl solution, the better passivation behaviour of SAC 305 was detected compared to SAC105 and Sn-3.0Ag solder alloys. It was considered that SAC305 corrosion behaviour is better than Sn-3.0Ag due to the copper content in SAC305, and at the same time, SAC 305 corrosion behaviour was considered better than SAC105 due to the relatively higher Ag content [53]. Furthermore, in aerated 0.1 M NaCl solution, the better corrosion behaviour of Sn-3.0Ag-3.0Cu was detected compared to SAC305. It was attributed to the increase in the Cu content with the same Ag percentage [60]. The ability of passivation when varying Ag content in Sn-$X$Ag-0.5Cu ($X$ = 1, 2, 3, 4 wt %) alloy was investigated in deaerated 3.5 wt % NaCl solution [51]; it has been found that the ability to passivate increases when the Ag content is beyond 2%. In aerated 0.1 M NaCl solution, the significantly higher corrosion resistance for Sn-2.9Ag-6.7Cu was measured compared to Sn-3.1Ag-0.8Cu solder alloy. It was also attributed to the increase in the copper content [76].

The ECC of commercial SAC305 solder alloy compared to SAC305 solder joints was investigated using the salt spray test in 5 wt % NaCl solution [77]; the SAC 305 solder joint is proved to be corroded easily compared to the commercial SA305 due to the presence of a Cu pad that promotes the dissolution of Sn into the corrosive medium.

The ECC of SAC105 and Sn-3.8Ag-0.7Cu solder alloys, isothermally aged at 120 °C for two different periods of 4 h and 72 h, was investigated in 0.5M NaCl aqueous solution with a measured

pH of ≈6.0 [78], $i_{corr}$ is decreasing in the following order SAC105-4 h > SAC387-4 h > SAC105-72 h > SAC387-72 h. For the same ageing time, SAC387 shows better corrosion resistance; for the same alloy, corrosion resistance increase with the ageing time.

The ECC of Sn-3.8Ag-0.7Cu and Sn-3.66Ag-0.91Cu alloys was investigated in the air-saturated aqueous solutions of NaCl (0.01, 0.1, 0.5, 1 M) [42], in which 95.5Sn-3.8Ag-0.7Cu shows a higher corrosion rate (based on $i_{corr}$) than 95.43Sn-3.66Ag-0.91Cu in all concentrations. Moreover, the ECC of the previous two SAC alloys was investigated in a solution containing $NO_3^-$, $SO_4^{-2}$, and $Cl^-$ ions of concentration equivalent to those in acid rains (pH = 3.63, 3.92, 4.15) [42], and it has been found that 95.43Sn-3.66Ag-0.91Cu alloy was more corrosion resistant than 95.5Sn-3.8Ag-0.7Cu, except for at pH = 3.63.

The ECC of SAC305 was investigated in 1 M HCl compared to 3.5 wt % NaCl solution [79], and SAC305 has worse corrosion behavior in 1 M HCl, since it shows much higher $i_{corr}$ and $i_p$ density ($i_{corr}$ = 39.83 μA, $i_p$ = 0.0506 A) than its values in 3.5 wt % NaCl solution ($i_{corr}$ = 2.19 μA, $i_p$ = 0.0105 A).

In addition to an NaCl environment, the ECC of SAC alloys in other electrolyte environments was investigated; SAC305 in 6 M KOH was seriously attacked, the primary passive film was broken down, and pitting corrosion occurred [73]. The ECC of SAC305 was investigated in aerated 1 M HCl solution [44]; it showed excellent corrosion resistance by having a stable passive film due to the relative higher presence of $SnO_2$ compared to SnO, and the test showed active, passive, and pseudopassivation regions: the values of the test are approximately $E_{corr}$ = 0.6 VSCE and $i_{corr}$ = 10 μA.

## 3.2. Corrosion Studies Related to the Alloying Elements on SAC Alloys

In addition to the variation of the compositional content within SAC alloy, the incorporation of alloy elements was investigated to evaluate the ECC behaviour of SAC alloys. The corrosion resistance of SAC305-*X*Al (*X* = 0, 0.1, 0.5 wt %) solder alloy decreased with the doping of aluminium in aerated 3.5 wt % NaCl solution [54]. However, it has been found that doping SAC105 with 0.2 wt % aluminium had the lowest corrosion rate among SAC105-*X*Al (*X* = 0, 0.2, 0.5, 1 wt %) solder alloy in 5 wt % NaCl solution [57].

Doping SAC305 with germanium accelerated the corrosion in 3 wt % NaCl solution [80], among SAC305-*X*Ge (*X* = 0, 0.2, 0.5, 1, 2, 5, 8 wt %), doping with 1 wt % had the lowest anti-corrosive capacity by showing the lowest noble value of $E_{corr}$ and the highest value of $i_{corr}$. SAC305 was doped within the following three cases: (1) only germanium, (2) only zinc, and (3) a mixture of both elements. The doping content percentages were the following (0, 0.2, 0.5, 1, 2, 5, 8 wt %) in 3.5 wt % NaCl solution in each case [81], which resulted in the following main observations: (1) germanium accelerated ECC with the highest $i_{corr}$ accompanied with 1 wt %; (2) zinc accelerated ECC with the highest $i_{corr}$ accompanied with 1 wt %; (3) a mixture of both elements accelerated ECC with the highest $i_{corr}$ accompanied with 1 wt %; and (4) the highest $i_{corr}$ among all the three cases was accompanied with 1 wt % zinc.

The corrosion resistance of SAC105-*X*Zn (*X* = 0, 0.1, 0.5, 1 wt %) solder alloy decreased with the doping of zinc in 3.5 wt % NaCl solution [55]. Super quality degradation and numerous small areas of material spalling were observed on SAC105-0.5Fe compared to SAC105 in 5 wt % NaCl solution [82].

SAC305 was doped within the following three cases: (1) only indium, (2) only zinc, and (3) a mixture of both elements. The doping content percentages were the following (0, 0.2, 0.5, 1, 2, 5, 8 wt %) in 3.5 wt % NaCl solution in each case [83], which resulted in the following main observations: (1) indium accelerated ECC with the highest $i_{corr}$ accompanied with 1 wt %; (2) zinc accelerated ECC with the highest $i_{corr}$ accompanied with 1 wt %; (3) a mixture of both elements accelerated ECC with the highest $i_{corr}$ accompanied with 2 wt %; and (4) the highest $i_{corr}$ among all the three cases was accompanied with 2 wt % zinc.

The ECC behaviour of SAC105-Fe-1Bi was compared to SAC105 and SAC105-Fe-2Bi, and the improved corrosion resistance was attributed to the lowest micro-galvanic coupling between Sn and $Ag_3Sn$ [59]. SAC105-0.05Ni and SAC105-1Ni showed better corrosion and passivation ability than

SAC105-0.5Ni due to the formation of tin oxyhydroxide chloride species on the surface of the solder joints [50].

### 3.3. Corrosion Studies of SAC Alloys Compared to Other Alloys

The ECC behaviour of SAC alloys was investigated compared to other lead-free and Sn-Pb solder alloys to evaluate their competence and efficiency. The ECC of Sn-0.7Cu, Sn-3.5Ag, Sn-0.3Ag-0.7Cu, and Sn-3.5Ag-0.7Cu was investigated in 3.5 wt % NaCl solution [72]; it has been found that $E_{corr}$ shifts to lower noble values and $i_{corr}$ decreases in the following order: pure Sn < Sn-3.5Ag-0.7Cu < Sn-0.3Ag-0.7Cu < Sn-3.5Ag < Sn-0.7Cu. The ECC behaviour of Sn-2.5Ag-0.5Cu and Sn-48Bi-2Zn was investigated in 3% NaCl solution [84]; Sn-2.5Ag-0.5Cu showed higher $R_{ct}$ than Sn-48Bi-2Zn. The ECC behaviour of Sn-9Zn, Sn-8Zn-3Bi, Sn-3.0Ag-0.5Cu, Sn-3.5Ag-0.5Cu-9In, and Sn-37Pb as a reference was investigated in 3.5% NaCl solution and 0.1% adipic acid solution [52]; it was found that SAC305 shows the best corrosion resistance (lowest $i_{corr}$ and highest $R_p$) in both solutions. However, Sn-3.5Ag-0.5Cu-9In still has better corrosion behaviour than Sn-9Zn and Sn-8Zn-3Bi in both solutions.

The ECC behaviour of Sn-3.5Ag-0.9Cu, Sn-3.5Ag, Sn-0.7Cu, Sn-57Bi, and Sn-9Zn compared to pure Sn and Sn-37Pb was investigated in aerated 0.1 M NaCl solution [69]; it has been found that Sn–3.5Ag-0.9Cu, Sn-3.5Ag, and Sn–0.7Cu solders have the best resistance to the corrosion. A corrosion rate 0.1 mm/year was suggested to be accepted, and based on this value, Sn-3.5Ag-0.9Cu is acceptable to be installed in a 0.1 M NaCl environment. The 94.5Sn-3.8Ag-1.5Cu alloy had a better ECC behaviour than 73.9Sn-23.1Pb in 0.1 M aerated NaCl solution [43], showing a larger pseudopassivation range, lower pseudopassivation current density, and higher corrosion resistance than 73.9Sn-23.1Pb. Using a salt spray test, SAC405 showed a higher degree of corrosion after 48 and 96 h of exposure than Sn-0.37Pb in 5% NaCl solution [74].

The ECC behaviour of Sn-3.5Ag, Sn-0.7Cu, Sn-3.8Ag-0.7Cu, and Sn-37Pb as a reference was investigated in 3.5 wt % NaCl solution [41], and it has been found that Sn-3.8Ag-0.7Cu has the lowest noble value of $E_{corr}$, while the corrosion resistance of Sn-Ag-Cu is similar to that of Sn–0.7Cu and Sn-3.5Ag, but that of Sn-3.5Ag is better than that of two others. Although SAC305 solder alloy showed a more noble value of $E_{corr}$ compared to Sn-8.5Zn-0.05Al-XGa (X = 0, 0.02, 0.05, 0.2 wt %) solder alloy in deaerated 3.5 wt % NaCl [85], it showed a much higher $i_{corr}$ and corrosion rate as well as a much lower $R_p$ than all other alloys. Furthermore, the value of the critical current density ($i_{crit}$) is much higher than $i_p$, indicating an increase in the corrosion.

### 3.4. Corrosion Studies of Micro-Alloyed SAC Alloys

The corrosion behaviour of four micro-alloyed lead-free solders [86] were investigated in 3.5 wt % NaCl solution: SAC0307-0.05Ni, SAC0807-0.1Bi, SAC0807-0.05Ni, SAC0807-3Bi-1.4Sb-0.15Ni, and SAC305 as a reference. According to the $i_{crit}$, $i_p$ values, and corrosion depth measurements, generally the low Ag content alloys show better corrosion behaviour than the reference, except for SAC3807-3Bi-1.4Sb-0.15Ni. It indicates that the bismuth and silver content may pose a relatively high corrosion risk. The worst corrosion resistance was detected in the case of SAC0807-3Bi-1.4Sb-0.15Ni. In the corrosion product, around 2.5 wt % of bismuth and a relatively huge amount of silver (8%) was found [87].

Investigation of the previous micro-alloyed solders was extended [88] by studying their corrosion behaviour in 0.38 wt % $MgCl_2$ solution. In the case of $MgCl_2$ solution, SAC0807-3Bi-1.4Sb-0.15Ni showed the highest ECC susceptibility according to $i_p$ and $i_{crit}$. It suggests the impact of bismuth and antimony. The effect of electrolyte concentration was studied in different NaCl concentrations (0.1 mM, 1 mM, 10 mM, 500 mM, and a saturated one) as well. The better corrosion properties have been reported for SAC305 in all the concentrations except at 500 mM NaCl concentration.

### 3.5. Corrosion Studies of Composite SAC Solder Alloys

Due to the recent development in nanotechnology, the doping of SAC solder alloys with different kinds of nanoparticles has become an available choice. SAC305 alloys reinforced with 0.05 wt % NiO, $Fe_2O_3$, and $TiO_2$ nanoparticles were investigated in 0.1 M NaCl solution [58] and compared to SAC 305 as the reference. Enhanced corrosion resistance was reported for all the doped solders compared to the reference, since the existence of nanoparticles as reinforced tiny particles in the solder paste retard the corrosion attack by the aggressive chloride ion. SAC305 solder paste alloyed with 0.3 wt % $Ag_3Sn$ and 0.3 wt % $Cu_3Sn$ nanoparticles respectively showed the best corrosion resistance among the additions ($X$ = 0, 0.3, 0.6, 0.9, 1.2 wt %) for both the nano-$Ag_3Sn$ and nano-$Cu_3Sn$ in 3.5 wt % NaCl solution. The corrosion products at 0.3 wt % alloying have the thinnest sheet structure compared to the sheets that are getting thicker as the amount of nanoparticles addition increases [48]. Moreover, it has been also found for Sn-$x$Ag-$y$Zn solder in NaCl solution [89] that the thin sheet structure is the corrosion product of β-Sn, while the thick one is formed by the corrosion of a eutectic structure. Since the eutectic structure has a large number of interfaces, the corrosion resistance will be decreased due to the defects at the interface. Alloying by $Al_2O_3$ nanoparticles (0.12 wt %) [66] refines the microstructure of SAC0307 and acts as a physical barrier to prevent further corrosion in 0.5 M NaCl solution. So, the corrosion mechanism was only controlled by the charge transfer, while in the case of the reference SAC0307, the corrosion process was controlled by the charge transfer and diffusion process as well.

Ni-coated carbon nanotubes (CNTs) ($X$ = 0, 0.01, 0.03, 0.05 wt %) were added to 95.8Sn-3.5Ag-0.7Cu [90] to refine the microstructure. In the corrosion test performed in 0.3% $Na_2SO_4$ solution, it has been found that the 0.01 wt % of nanotubes is well dispersed, and the passive film is less rough than in the higher doped alloys. Another study [91] showed that the addition of Ni-CNTs into SAC solder alloy results in the formation of $Ni_3Sn_4$, which is a stable phase serving as a barrier to oxygen diffusion at the solution/metal interface. For 95.8Sn-3.5Ag-0.7Cu doped with the following contents of Ni-CNTs ($X$ = 0, 0.01, 0.03, 0.07 wt %) in 3.5 wt % NaCl solution [56], it has been reported that the increase in Ni-CNTs content results in a decrease in the grain size, which enhanced the compactness of the passive layer. The role of Ni-CNTs is the reduction of the galvanic coupling between the Sn anode and $Ag_3Sn$ cathode.

The addition of graphene nanosheets (GNSs) with the following contents ($X$ = 0, 0.03, 0.07, 0.1 wt %) to SAC305 was investigated in 3.5 wt % NaCl solution [92]. The addition of GNSs resulted in enhanced corrosion resistance compared to the undoped solder alloy. The addition of GNSs provides a lower diffusion length by increasing the tortuosity of oxygen diffusion pathways. GNSs can also enhance the corrosion resistance in a similar way to Ni-CNTs by acting as a third electrode, reducing galvanic corrosion between the Sn anode and $Ag_3Sn$ cathode. However, when the content of GNSs increases above 0.03 wt %, there is a decrease in the corrosion resistance, since GNSs tends to stack together due to van der Waals forces.

### 3.6. Recently Developed Permanent Magnet Stirring

The enhancement of SAC solder alloy corrosion resistance can also be achieved during the solidification process, for example applying the Permanent Magnet Stirring (PMS) method. SAC205 corrosion was investigated in 3.5 wt % NaCl solution [65]; the higher corrosion rate before applying PMS was attributed to the coarse $Ag_3Sn$ and $Cu_6Sn_5$ IMC precipitates between the Sn-rich phase and the IMCs. Moreover, grains consisted of single-crystal morphology with $(111)_{Sn}$ facets had higher susceptibility to the pitting attack, and the application of PMS refined the microstructure and transformed it from lamellar grains into a homogeneous equiaxed grain structure, which resulted in enhancement of the passivation of the alloy matrix. The PMS effect on corrosion behaviour is still very rare in the literature. PMS developed as an alternative approach of conventional Electromagnetic Stirring (EMS) has been also reviewed and published elsewhere [93] with the simulation modelling, experimental results, and industrial application.

### 3.7. Temperature and Cooling Rate Effect on the Corrosion SAC Alloys

The cooling rate is a very important factor in the formation of $Ag_3Sn$ plates during the reflow soldering. They are only generated in the middle period of the cooling stage [94]. The effect of the cooling process (commercial, air-cooled, and furnace-cooled) on SAC305 corrosion was investigated at different temperatures (25, 45, 65 °C) in 3.5 wt % NaCl environment [49]. The decrease in the cooling rate resulted in a significant change on the microstructure of SAC305 solder alloy, and there was a change in $Ag_3Sn$ morphology from fine-fibre to large platelet. All the previous three SAC alloys showed passive behaviour at 25 °C, while at higher temperatures, they remain active. Probably, this is due to the low area ratio of Sn to $Ag_3Sn$ phase, as for air-cooled and furnace-cooled samples, pits were distributed throughout the surface after anodic polarisation. The commercial SAC305 showed better corrosion characteristics compared to the furnace-cooled and air-cooled SAC305 solders by having a more compact and adherent surface film of corrosion products.

Studies have also been done under accelerated conditions, in high-temperature and high-humidity environments 75 °C/100% RH for SAC305 [95]. Commercial SAC305 showed better corrosion behaviour compared to the furnace-cooled and air-cooled SAC305 solders due to the oxide film, which consists of $SnO_2$ and SnO protecting the $Ag_3Sn$ particles at the outer layer of the film. These results are in agreement with another similar study for the three same types of alloys in 60 °C/100% RH [96].

Tables 2 and 3 summarise the corrosion behaviour of SAC solder alloys in terms of the most important corrosion parameters.

**Table 2.** Experimental data of SAC alloys after Linear Sweep Voltammetry (LSV) in 3.5 wt % NaCl solution.

| Type of SAC | $E_{corr}$ (mV) | $i_{corr}$ (mA/cm$^2$) | Reference |
|---|---|---|---|
| SAC305 | −415 (SCE) | $7.175 \times 10^{-5}$ | [81] |
| SAC305-0.2Ge | −499 (SCE) | $2.213 \times 10^{-4}$ | [81] |
| SAC305-0.5Ge | −593 (SCE) | $2.842 \times 10^{-4}$ | [81] |
| SAC305-1Ge | −661 (SCE) | $1.181 \times 10^{-3}$ | [81] |
| SAC305-2Ge | −598 (SCE) | $3.089 \times 10^{-4}$ | [81] |
| SAC305-5Ge | −572 (SCE) | $3.859 \times 10^{-4}$ | [81] |
| SAC305-8Ge | −545 (SCE) | $2.539 \times 10^{-4}$ | [81] |
| SAC305 | −524 (KCl-Ag/AgCl) | 0.154 | [54] |
| SAC305-0.1Al | −649 (KCl-Ag/AgCl) | $0.54 \times 10^{-3}$ | [54] |
| SAC305-0.5Al | −719 (KCl-Ag/AgCl) | $1.07 \times 10^{-3}$ | [54] |
| SAC305 | −524 (Ag/AgCl saturated KCl) | $27.8 \times 10^{-3}$ | [53] |
| SAC105 | −510 (Ag/AgCl saturated KCl) | $318 \times 10^{-3}$ | [53] |
| SAC305 | −415 (SCE) | $7.175 \times 10^{-5}$ | [83] |
| SAC0307 | −840 (saturated Ag/AgCl) | $0.83 \times 10^{-3}$ | [72] |
| Sn-3.5Ag-0.7Cu | −837 (saturated Ag/AgCl) | $1.03 \times 10^{-3}$ | [72] |
| Commerical SAC305 | −877 (SCE) | $9.4 \times 10^{-3}$ | [49] |
| Air-cooled SAC305 | −867 (SCE) | $9.6 \times 10^{-3}$ | [49] |
| Furnace-cooled SAC305 | −885 (SCE) | $9.5 \times 10^{-3}$ | [49] |
| SAC305 | −605 (SCE) | $5.37 \times 10^{-4}$ | [52] |
| Sn-3.5Ag-0.5Cu-9In | −578 (SCE) | $7.413 \times 10^{-3}$ | [52] |
| SAC105 | −668 (SCE) | $0.451 \times 10^{-3}$ | [55] |
| SAC105-0.1Zn | −676 (SCE) | $0.705 \times 10^{-3}$ | [55] |
| SAC105-0.5Zn | −696 (SCE) | $0.891 \times 10^{-3}$ | [55] |
| SAC105-1Zn | −795 (SCE) | $9.720 \times 10^{-3}$ | [55] |
| SAC305 | −1052 (SCE) | $2.19 \times 10^{-3}$ | [79] |
| SAC305 | −1078 (SCE) | $5.2 \times 10^{-3}$ | [97] |
| SAC305 | −410 (SCE) | $1.469 \times 10^{-2}$ | [48] |

**Table 2.** *Cont.*

| Type of SAC | $E_{corr}$ (mV) | $i_{corr}$ (mA/cm$^2$) | Reference |
|---|---|---|---|
| SAC105 | −810 (Ag/AgCl) | 4.6 | [51] |
| SAC205 | −814 (Ag/AgCl) | 6.06 | [51] |
| SAC305 | −804 (Ag/AgCl) | 3.14 | [51] |
| SAC405 | −636 (Ag/AgCl) | 2.25 | [51] |
| Sn-3.8Ag-0.7Cu | −727 (SCE) | 0.089 | [41] |
| SAC205 | −989 (Ag/AgCl) | $114 \times 10^{-3}$ | [65] |
| SAC205 soldified with PMS | −652 (Ag/AgCl) | $31 \times 10^{-3}$ | [65] |
| Sn-3.5Ag-0.7Cu | −1220 (SCE) | - | [56] |
| Sn-3.5Ag-0.7Cu/0.01 Ni-CNTS | −820 (SCE) | - | [56] |
| Sn-3.5Ag-0.7Cu/0.03 Ni-CNTS | −780 (SCE) | - | [56] |
| Sn-3.5Ag-0.7Cu/0.07 Ni-CNTS | −480 (SCE) | - | [56] |
| SAC305 | −590 (SCE) | $0.14 \times 10^{-3}$ | [86] |
| SAC0307-0.05Ni | −569 (SCE) | $0.083 \times 10^{-3}$ | [86] |
| SAC0807-0.1Bi | −597 (SCE) | $0.011 \times 10^{-3}$ | [86] |
| SAC0807-0.05Ni | −568 (SCE) | $0.024 \times 10^{-3}$ | [86] |
| SAC0807-3Bi-1.4Sb-0.15Ni | −570 (SCE) | $0.84 \times 10^{-3}$ | [86] |

**Table 3.** Experimental data of SAC alloys after LSV in other solutions.

| Type of SAC | Type and Concentration of the Solution | $E_{corr}$ (mV) | $i_{corr}$ (mA/cm$^2$) | Reference |
|---|---|---|---|---|
| SAC305 | 3% NaCl | −415 (SCE) | $7.175 \times 10^{-5}$ | [80] |
| SAC105 | 5 wt % NaCl | −741.6 (SCE) | $70.8 \times 10^{-3}$ | [57] |
| SAC305 | 0.1 M NaCl | −178 | $12.64 \times 10^{-6}$ | [58] |
| SAC305-0.05Ni | 0.1 M NaCl | −535 | $1.604 \times 10^{-6}$ | [58] |
| SAC305-0.05Fe$_2$O$_3$ | 0.1 M NaCl | −309 | $1.161 \times 10^{-6}$ | [58] |
| SAC305-0.05TiO$_2$ | 0.1 M NaCl | −278 | $1.113 \times 10^{-6}$ | [58] |
| SAC305 | 0.1 M NaCl | −520 (SCE) | - | [60] |
| Sn-3.0Ag-3.0Cu | 0.1 M NaCl | −395 (SCE) | - | [60] |
| Sn-3.1Ag-0.8Cu | 0.1 M NaCl | −670 (SCE) | - | [76] |
| Sn-2.9Ag-6.7Cu | 0.1 M NaCl | −545 (SCE) | - | [76] |
| SAC205 | 3% NaCl | −610 (Ag/AgCl) | - | [84] |
| SAC305 | 6 M KOH | −1108 (Hg/HgO) | 0.1795 | [73] |
| SAC305 | 1 M HCl | −600 (SCE) | 0.01 | [44] |
| SAC305 | 1M HCl | −713.3 (SCE) | $39.83 \times 10^{-3}$ | [79] |
| SAC105 | 0.1 M NaCl | −599 (SCE) | $0.145 \times 10^{-3}$ | [59] |
| SAC105-Fe-1Bi | 0.1 M NaCl | −538 (SCE) | $0.128 \times 10^{-3}$ | [59] |
| SAC105-Fe-2Bi | 0.1 M NaCl | −577 (SCE) | $0.178 \times 10^{-3}$ | [59] |
| SAC0307 | 0.5 M NaCl | −750 (SCE) | $3.47 \times 10^{-3}$ | [66] |
| SAC0307-0.12Al$_2$O$_3$ | 0.5 M NaCl | −680 (SCE) | $1.15 \times 10^{-3}$ | [66] |
| Sn-3.5Ag-0.7Cu | 0.3% Na$_2$SO$_4$ | −685.92 (SCE) | $3.6252 \times 10^{-4}$ | [90] |
| Sn-3.5Ag-0.7Cu/0.01 Ni-CNTs | 0.3% Na$_2$SO$_4$ | −662.08 (SCE) | $2.3297 \times 10^{-4}$ | [90] |
| Sn-3.5Ag-0.7Cu/0.03 Ni-CNTs | 0.3% Na$_2$SO$_4$ | −661.59 (SCE) | $12.302 \times 10^{-4}$ | [90] |
| Sn-3.5Ag-0.7Cu/0.05 Ni-CNTs | 0.3% Na$_2$SO$_4$ | −743.37 (SCE) | $10.462 \times 10^{-4}$ | [90] |
| SAC105 aged for 4 h at 120 °C | 0.5 M NaCl | −497 (Ag/AgCl 1 M KCl) | $18.45 \times 10^{-3}$ | [78] |
| SAC105 aged for 72 h at 120 °C | 0.5 M NaCl | −654 (Ag/AgCl 1 M KCl) | $1.99 \times 10^{-3}$ | [78] |
| SAC387 aged for 4 h at 120 °C | 0.5 M NaCl | −475 (Ag/AgCl 1 M KCl) | $17.48 \times 10^{-3}$ | [78] |
| SAC387 aged for 72 h at 120 °C | 0.5 M NaCl | −585 (Ag/AgCl 1 M KCl) | $0.37 \times 10^{-3}$ | [78] |
| Sn-3.8Ag-1.5Cu | 0.1 M NaCl | −390 (SCE) | 7 | [43] |

## 4. Effect of Fluxes

In soldering technology, the flux is used to remove the metal oxides that natively exist on the surface. Two major types of fluxes are used: water-soluble fluxes and alcohol-based fluxes. Nowadays, the so-called no-clean fluxes are widely used. No-clean flux consists of a solvent (hydrocarbons, ethers, alcohols, etc.), activator (mainly weak organic acids (WOAs), or halide compounds), and additives (inhibitors, dyes, plasticisers etc.). They need to be easily removed/decomposed at soldering temperatures [98] during the automated (wave, reflow, or selective) soldering processes.

Reflow and wave soldering are susceptible to the corrosion resulted from flux residues [99]. However, the flux residue effect is obvious in hand soldering, as less efficient soldering is still

unavoidable in some cases such as mounting add-on devices [100] or in the case of space electronics. During the hand soldering process, having an uncontrolled and excessive amount of flux (and also the temperature is much higher than it is in wave soldering) and an uneven temperature results in unreacted flux residues causes a large increase in cathodic and anodic reactivity due to the increased probability of water layer formation as a conductive path.

The aggressiveness of flux residue is a function of its pH value (acidity) and the applied temperature. It has been reported [98] in the case of succinic, adipic, and glutaric acid that there is a significant level of WOA residues indicating an incomplete decomposition of the fluxes within the temperature profile of the soldering process. Humidity tests (60–98% RH) showed that glutaric acid is causing the highest susceptibility to corrosion, followed by succinic acid and adipic acid, respectively. Another similar study [101] introduced different flux systems containing different types of WOA, acid numbers, and solid contents. It has been shown that the solder flux systems can be used until a critical value of RH %, above which the corrosion properties are worse.

Furthermore, it has been found that improper rinses can increase the contamination of PCB after soldering with no-clean solder paste [102]. In addition, it has proven that no-clean flux can be mainly responsible for the failure of PCB [103] by providing the majority of ionic bromides residues that generated from the pyrolytic dissociation of covalent bromides in flux under high-soldering temperature. Therefore, tested as-received no-clean flux using the conventional test methods can be wrongly considered as "halide-free"; hence, cleaning is essential when using "highly covalently bound halids" no-cleaning flux.

The flux residue is an important aspect of the electrochemical migration (ECM) topic [104–107], which is a humidity-induced failure and considered a type of corrosion that imposes a high risk for the reliability of the electronics. In these cases, the metal is dissolved in ionic form from its initial location, migrated via the electrolyte in the presence of bias voltage, and deposited at the cathode side. With time, it causes a continuous degradation (due to dendrite growth) of surface insulation resistance and may lead to catastrophic failure in the electronics.

## 5. Conclusions

In this work, different SAC alloys corrosion studies were reviewed with the mostly applied electrochemical techniques, and the results were summarised. The main messages are the following:

- LSV is the most frequently applied technique to determine the corrosion parameters. Simple evaluation is possible; for example, the more noble values of $E_{corr}$ can indicate a better corrosion behaviour and $i_{corr}$ is a reliable parameter to evaluate the corrosion rate. Although this technique is widely used, it is not recommended for evaluating long-term corrosion performance.
- EIS is the most reliable technique; it evaluates in more detail the resistance of the passive film and the charge transfer resistance at the passive film/metal interface, and it also provides information about the electrolyte layer formation as well.
- Both techniques are also useful for pitting corrosion detection, which is one of the typical types of the solder alloy corrosion processes.
- Within the group of SAC alloys, it is important to study the variation of Ag and Cu content, as these elements are showing potential corrosion accelerating effects.
- Composite solders have great potential to enhance the corrosion performance of SAC alloys. However, this field is not deeply addressed in the literature.
- Slow cooling rates show a significant effect on the grain size, which is directly affecting the corrosion resistance of SAC alloys. The corrosion resistance of solders is expected to decrease when operating under high temperatures and high humidity environments.
- The structure-based corrosion resistance of SAC alloys can be further enhanced by technological improvements, such as the application of PMS during the solidification process.

- Flux residues may impose a high risk of the corrosion even in the presence of no-clean flux, and the risk becomes very high in case of hand soldering applications.

**Author Contributions:** Conceptualisation: B.M.; writing—original draft preparation: A.G.; writing—review and editing: I.F., Z.K., G.H., B.I., B.M. All authors have read and agreed to the published version of the manuscript.

**Funding:** This research received no external funding.

**Acknowledgments:** The research reported in this paper was supported by the Higher Education Excellence Program of the Ministry of Innovation and Technology in the frame of Nanotechnology and Material Science research area of Budapest University of Technology and Economics (BME FIKP-NAT), in the frame of the National Research. The work is also supported by the Pro Progressio Foundation (BME) and by the National Research, Development and Innovation Office–NKFIH, FK 132186.

**Conflicts of Interest:** The authors declare no conflict of interest.

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
