# Peer review of "Electrochemical Corrosion of SAC Alloys: A Review"

_metals, doi:10.3390/met10101276_

Round 1

Reviewer 1 Report

Report on the manuscript “ Electrochemical corrosion of SAC alloys: a review”, AR. Gharaibeh et al.

Ref. metals-927014.

General comments: 

The manuscript presents a review on studies dealing with the electrochemical corrosion of tin-silver-copper (SAC) solder alloys. The manuscript presents a review on an interesting topic but offers an excessively concise view of the results when compared with the relatively extensive description of analytical techniques. Major revision is recommended based on the following considerations.

General remarks:

  1. I) The description of electrochemical techniques is limited to potentiodynamic curves and EIS. No other techniques (electrochemical noise, SECM, for instance) have been applied to SACs?
  2. II) The authors provide a unique example of “typical” potentiodynamic curve in Fig. 1, but the logical practice when revising literature in section 3 is to provide other representative examples of possibly existing different behaviors.

III) The above consideration also applies for EIS measurements, limited to a unique example in Figs. 2 and 3. Are all impedance spectra measured at the open circuit potential? Taking into account the relatively complex behavior in Fig. 1, probably EIS measurements at different bias potential are of interest and have been studied in literature.

  1. IV) The equivalent circuit in Fig. 4 does not correspond to the EIS in Figs. 2 and 3 and the explanation of the meaning of the different elements in the text is not entirely clear.
  2. V) Presumably, there are different equivalent circuits in literature used to describe the response of SACs to corrosion tests. A representative sample of used circuits should be presented.
  3. VI) Table 1 contains, apparently (no clear heading is provided), the standard potentials of the different metal couples involved in SACs. The representativity of such data is, however, uncertain because these potentials are sensitive to the electrolyte, in particular (in regard to corrosion tests) to the presence of chloride (complexing) ions.

Author Response

Dear Reviewer,

We are truly grateful for your critical comments and thoughtful suggestions. Based on these comments and suggestions, we have made careful modifications to the original manuscript. All changes/answers made to the text are highlighted with a yellow background. You will find it as an attachment.

Kind Regards

Bálint Medgyes

Reviewer 2 Report

The article is well systematized, considers a large amount of material, is understandable. The publication of this material is of interest both to specialists directly in the field of Tin-silver-copper (SAC) solder alloys and their corrosion, and in related fields.

Author Response

Dear Reviewer,

Thank you very much for Your Comments! It is highly appreciated.

Kind Regards

Dr. Bálint Medgyes

Assoc Prof at BME

Reviewer 3 Report

The problem of safe soldering without the use of lead is urgent. The tin-silver-copper system can be a good alternative to the tin-lead solders. In their search for the best composition, researchers have written over a hundred publications on the SAC system. New researchers need to read a large number of articles to study the issue. It is difficult for business representatives to choose the best alloy from all the variety of results. Therefore, the proposed work is an important step towards the systematization of knowledge in this area. The method for determining the corrosion resistance of the SAC system is shown. Selected specific corrosion indicators for comparison. A review of 104 literature sources has been carried out. However, figuring out which alloy is best in a particular environment is still a challenge. I would like to wish the authors to move on to the second step of this complex task - to translate data on the corrosion resistance of different systems from a textual form to a graphical form, to indicate the relationship between the content of elements, the working environment, and corrosion indicators. This will make it easier to compare the corrosion resistance of the alloys.

Author Response

Dear Reviewer,

We are truly grateful for your critical comments and thoughtful suggestions. Based on these comments and suggestions, we have made careful modifications to the original manuscript.

We have tried to translate data on the corrosion resistance of different systems from a textual form to a graphical form. The important parameters such as Ecorr, Icorr are collected in Table 2 and 3. We hope It will be acceptable for you. Please, see the revised manuscript.

Kind Regards

Dr. Bálint Medgyes

Assoc. Prof. at BME

Round 2

Reviewer 1 Report

The manuscript can be published in its current version